# OpenReview forum: "RT-Trajectory: Robotic Task Generalization via Hindsight Trajectory Sketches"
_ICLR.cc/2024/Conference — ICLR 2024 spotlight_

### Official Review · Reviewer_YmAB · 2023-10-30

**Soundness:** 2 fair
**Presentation:** 3 good
**Contribution:** 2 fair
**Rating:** 5
**Confidence:** 4

**Summary:**

This paper proposes an approach to task specification based on providing, at test time, an approximate trajectory to accomplish the task. More specifically, the trajectory is provided as a 2D image, representing the projection on the camera of the end-effector position, the interaction points (where the gripper should be opened and closed), and possibly the height of the end effector. I see the approach as an instantiation of the general problem of trajectory tracking under constraints. The main contribution is adapting this idea to make it compatible with sensorimotor models and imitation learning. The approach is evaluated over multiple experiments in the real world, where it outperforms baselines without test-time goal conditioning and end-goal conditioning only.

**Strengths:**

The main strengths of the paper are:
1. The proposed approach, despite being a small extension of the well-known body of literature on trajectory tracking, is original. Using a trajectory to define the task at test time is the default for all control systems. The fact that the trajectory is not feasible is generally not a problem for feedback systems (See Borrelli et al., Predictive control for linear and hybrid systems). However, representing such a trajectory as an image and conditioning a model on it is an interesting idea.
2. While I don't agree with some of the experimental setup (more on this below), I found the analysis on trajectory conditioning (Sec 4.3) and measuring generalization (Sec. 4.4) to be quite interesting and well executed.
3. The paper writing is very good and easy to follow. The figures help in understanding the core concepts.

**Weaknesses:**

The main weaknesses I see are:
1. The paper does not make a strong enough connection between the traditional idea of trajectory tracking and the proposed approach. For example, there is no section in the related work covering the history of this technique and its recent applications to robot learning. As mentioned above, I still see some innovations in the proposed approach (sensorimotor controllers are generally conditioned on a low-dimensional trajectory, not an image of it)—however, this lack of perspective results in overclaiming the system's novelty.
2. The comparison against baselines in Sec 4.2 is not apples to apples: while the proposed approach gets an input a trajectory on __how__ to complete the task, the other methods only access a higher level instruction (e.g. language), or the final goal represented as an image. It is, therefore, not surprising that with more and cleaner information, the proposed approach outperforms the baselines. I find the experiments in Table I to be much more informative. They show that when the trajectory is very coarse, e.g. when drawn by a human, the proposed approach can track it better than the off-the-shelf IK. However, these results also show that the approach is, on average, worse than generating a trajectory with a VLM and tracking it with an IK, which somehow is against the main paper's contribution.
3. Each subsection seems to evaluate on different "tasks", and it is unclear how the task selection is done. For example, why change the task from 4.2 to 4.3? And why even change it in the comparisons of Table 1? Note that my point is not that the task were handpicked, but rather that this dynamics makes drawing insights from experiments challenging.
4. The paper has as its main claim generalization. Still, its main experiments on generalization over scenes and tasks only occupy a small part of the paper (Sec. 4.5). At the very least, this part should become quantitative since this is what would back up the claims on generalization.

Some rather minor limitations:
1. The ordering of Fig. 2 seems rather arbitrary. For example, in my subjective opinion, I see trajectory and code to be much closer than what is shown right now. Therefore, either more justification is needed, or the figure should be removed.
2. The approach makes strong assumptions that contradict the claim on scalability. Specifically, the two main assumptions are: (1) the camera should be static at training and test time, as well as fully calibrated, and (2) the end effector should always be visible in the camera. This makes data collection and evaluation very challenging and limits the set of tasks that can be done.
3. In Fig. 8, I don't get what "unrelated semantic categories" mean for a low-level skill. At the level of trajectory, why is generalization over semantics a question? Isn't it obvious due to the abstraction of the skill? Or did I misunderstand the meaning?

**Questions:**

The main actions to improve the paper are in my opinion:

1. Better positioning of the work with respect to previous literature.
2. Motivation behind task selection
3. Quantitative experiments in the setup of Sec. 4.5

---

> ### Author Response · Authors · 2023-11-19
> **Response to Reviewer YmAB**
>
> Thank you for the appreciation of our work and the helpful comments. We have carefully incorporated improvements to our work based on your feedback. We hope we have addressed your concerns, and are happy to follow up with any additional questions!
>
> ---
>
> > Lack of strong enough connection between the traditional idea of trajectory tracking and the proposed approach
>
> We thank the reviewer for pointing out the missing discussion on traditional trajectory tracking for robotic control. We have included a dedicated subsection in the related work to situate our method against the relevant literature on trajectory tracking in control theory and robotics. Namely, we would like to highlight the body of work on adaptive and learning-based control which may be applied to closed-loop control for tracking reference trajectories, which may be provided after an initial trajectory planning stage or be updated dynamically in an online fashion. Most importantly, we emphasize that these model-based trajectory tracking algorithms are generally sensitive to accurate trajectory plans (which must be overspecified in ground truth state space) and are not robust to tracking errors in presence of dynamic or uncertain situations. Our proposed method makes fewer assumptions on the fidelity and accuracy of the coarse trajectory sketch, and aims to leverage the benefits of end-to-end learning to generalize to uncertain or complex scenarios. Nonetheless, we view our method and similar approaches as a hybrid abstraction that aims to leverage some of the benefits showcased in model-based trajectory tracking systems while also exploiting the benefits of end-to-end data-driven policy learning.
>
> ---
>
> > “The comparison against baselines in Sec 4.2 is not apples to apples.”
>
> We acknowledge that the comparison in Sec 4.2 is not apples to apples. The objective of this section is to demonstrate that trajectory-conditioned policies outperform baselines with language or goal image conditioning. It indicates a policy conditioning representation showing how to complete a task can be more amenable to task generalization than common baseline approaches.
>
> ---
>
> > “The approach is, on average, worse than generating a trajectory with a VLM and tracking it with an IK, which somehow is against the main paper's contribution.”
>
> Our approach performs worse on the task, Open Drawer, since the waypoints (always grasping the drawer handle) generated by LLM on this task are quite different from the motions in our training data (which mainly grasps the drawer edge). As shown in the Appendix (retry behaviors), RT-Trajectory can follow the trajectory in order to grasp the handle first, while finally opening the drawer by grasping the edge. Such retry behaviors are inherently impossible for the IK planner. Besides, we would like to point out that the IK planner requires relatively accurate trajectories of 6D poses while our approach only takes a 2D sketch as input.
>
> ---
>
> > “Each subsection seems to evaluate on different "tasks", and it is unclear how the task selection is done.”
>
> Thank you for pointing out this question about task selection. We have done additional experiments to compare RT-Trajectory against the IK planner given trajectories generated from human demonstrations videos on the Open Drawer skill and prompting LLMs on the Fold Towel skill, in order to ensure the same set of skills are evaluated in Table 1. Notably, the IK planner consistently fails to grasp the drawer edge (resulting in a success rate of 0), likely caused by the noisy trajectories generated from human demonstration videos for Open Drawer.
>
> | Method       | Open Drawer (human demo video) | Fold Towel (Prompting LLMs) |
> |--------------|--------------------------------|-----------------------------|
> | IK Planner   | 0%                             | 25%                         |
> | RT-Traj 2D   | 60%                            | 0%                          |
> | RT-Traj 2.5D | 90%                            | 25%                         |
>
> The reason why we only evaluate some of the seen or unseen skills in Sec 4.3 rather than all 7 unseen skills in Sec 4.2 is that the proposed unseen skills might be challenging for certain trajectory generation methods. For example, prompting LLMs to achieve swiveling a chair or folding a towel is empirically very difficult and out of the scope.
>
> ---
>
> > “Still, its main experiments on generalization over scenes and tasks only occupy a small part of the paper (Sec. 4.5).”
>
> In this work, we emphasize task-level generalization, and mainly present quantitative results on 7 unseen skills in Sec 4.2, as well as motion analysis in Sec. 4.4. Sec. 4.5. mainly shows some emergent abilities and case studies qualitatively.

---

> ### Author Response · Authors · 2023-11-19
> **Response to Reviewer YmAB (cont.)**
>
> > “The ordering of Fig. 2 seems rather arbitrary.”
>
> Thank you for the feedback. We agree that the figure can be subjective. We will remove it in the revision.
>
> ---
>
> > “The approach makes strong assumptions that contradict the claim on scalability.”
>
> Thank you for pointing out our assumptions that might affect scalability.
>
> 1) Static and calibrated camera during training and inference: When creating hindsight trajectory labels during training, we extracted segments from demonstrations which only contained stationary manipulation. During inference, one potential way to extend to mobile camera scenarios is to redraw trajectories if the robot moves during the rollout. As a future work, we are exploring automatically re-generating new trajectories on-the-fly during inference. In addition, we would like to emphasize that both our training data and evaluation are collected and conducted across different robots (distributing collection and evaluation on more than 10 robots). Thus, it is not required to have a highly precise camera calibration.
>
> 2) “The end effector should always be visible in the camera”: The end-effector is not required to always be visible within one episode. We will snap the projected trajectory to the edge of the image if the end-effector is out of view. It is also reasonable to assume that the end-effector is within the view for most time for a visual servoing system.
>
> ---
>
> > “what "unrelated semantic categories" mean for a low-level skill.”
>
> Thank you for the question. We have refined Fig. 8 and removed this description in the revision. For a learning-based policy, the task specification (e.g., semantics) can be highly related to the low-level motion. For example, for a language-conditioned policy, even if the low-level motion is similar, due to large gaps of language conditioning between training and test, the policy can still fail to generalize. We intend to emphasize that our approach can generalize to new tasks with unseen semantics because it is designed to generalize based on motion.

---

> > ### Comment · Reviewer_YmAB · 2023-11-22
> > **Thanks for your rebuttal**
> >
> > ```
> >  Most importantly, we emphasize that these model-based trajectory tracking algorithms are generally sensitive to accurate trajectory plans (which must be overspecified in ground truth state space) and are not robust to tracking errors in presence of dynamic or uncertain situations.
> > ```
> > This general statement is incorrect. There are several works that show how to only use an unfeasible trajectory by doing real-time replanning (e.g., MPPI). I would just focus on the fact that the trajectory is a sketch instead of a set of states.
> >
> > ```
> > We intend to emphasize that our approach can generalize to new tasks with unseen semantics because it is designed to generalize based on motion.
> > ```
> > This comes automatically by switching to trajectory tracking, I don't see how this can be a contribution of this paper. Isn't it obvious that this would happen by switching to trajectory tracking? Or am I missing something?
> >
> > ```
> >  It is also reasonable to assume that the end-effector is within the view for most time for a visual servoing system.
> > ```
> > For visual servo it does, but not in general for manipulation. I would make this clear in the revision.
> >
> > Interestingly, the trajectory generated by LLM prompting outperforms the proposed approach. Why does this happen? I would evaluate this more broadly and on all tasks of Fig. 4. If that works best overall, this might be even more scalable (no need for humans in the loop!).
> >
> > Finally, I see the point the paper makes a generalization, but I don't understand why one cannot do a more in-depth quantitative evaluation, possibly on the same tasks as before but with a different background/scene/drawer, etc.
> >
> > Overall, I maintain my opinion on the score. I think this is an interesting contribution, but I wish a more in-depth evaluation will be done to back up the generalization claims.

---

> > > ### Author Response · Authors · 2023-11-23
> > > **Thanks for the feedback**
> > >
> > > We appreciate the thoughtful comments, and have further improved our paper based upon your feedback. If these updates have addressed your concerns, we would appreciate an updated score that reflects the improvements to the submission.
> > >
> > > ```
> > > I would just focus on the fact that the trajectory is a sketch instead of a set of states.
> > > ```
> > >
> > > Thanks for this feedback, we agree and clarify that our response was mainly to highlight that our method does not assume trajectories given in ground-truth state space (irrespective of the accuracy/feasibility of the trajectory). We have updated the manuscript and citations to note that existing methods like MPPI may be able to handle inaccurate/infeasible trajectory targets in dynamic situations, and that our main contribution is to remove the over-specification assumption of trajectories by providing them in image space. Thanks for this suggestion.
> > >
> > > ```
> > > This comes automatically by switching to trajectory tracking, I don't see how this can be a contribution of this paper. Isn't it obvious that this would happen by switching to trajectory tracking? Or am I missing something?
> > > ```
> > >
> > > We agree that trajectory tracking methods also would see "motion generalization", but we would like to note that the premise of training vs. test "generalization" is ill-defined for model-based methods which do not have training vs. inference phases that data-driven imitation learning methods have. Our response was meant to clarify that our claims around generalizing to new motions with unseen semantics is only meaningful in the regime of data-driven methods.
> > >
> > > We also agree with the strong observation that such benefits of trajectory conditioning are "obvious" -- which was indeed the driving inspiration of our method. Put another way, our motivation was that trajectory sketches would enable the ability to generalize to both motions (thanks to the trajectory modality) as well as semantics/visuals (thanks to the end-to-end data driven training). Nonetheless to our best knowledge, our work is the first to introduce conditioning end-to-end imitation learning policies on coarse trajectory sketches (either by hindsight trajectories, human drawings, or automatically produced sketches). Thus, even if the benefits are well-motivated or obvious, the fact that such a simple idea was not explored before does make our work a valuable and original contribution.
> > >
> > > As an aside, we are optimistic that our work can provide an alternative perspective to how the many powerful benefits of optimal control approaches (which have been astutely pointed out by the reviewer) may be studied in conjunction with end-to-end robot learning methods (at the minimum via design philosophy and motivation).
> > >
> > > ```
> > > For visual servo it does, but not in general for manipulation. I would make this clear in the revision.
> > > ```
> > >
> > > Thanks for the feedback. We have clarified it in the limitation section.

---

> ### Author Response · Authors · 2023-11-21
> **Follow-Up to Reviewer YmAB**
>
> With the rebuttal period coming to a close, we hope we've addressed your concerns. We kindly ask that you consider raising your rating based on the above response. Feel free to let us know if you'd like us to address anything further. Thanks again for your thoughtful feedback!

---

> ### Author Response · Authors · 2023-11-23
> **Thanks for the feedback (cont.)**
>
> ```
> Interestingly, the trajectory generated by LLM prompting outperforms the proposed approach. Why does this happen? I would evaluate this more broadly and on all tasks of Fig. 4. If that works best overall, this might be even more scalable (no need for humans in the loop!).
> ```
>
> While it is true that the LLM-generated trajectories perform well for the "Pick" skill when paired with either the IK Planner Policy or RT-Trajectory Policy, we find that when using trajectories generated by LLMs, the IK Planner is roughly on par with our method (2.5D version) on average (IK Planner average success rate is 60%, while ours (2.5D) average success rate is 58%). In addition, executing the IK Planner with trajectories generated by prompting LLMs is not always the best across the board. In Table 1, we find that executing RT-Trajectory (2.5D) with trajectories generated by human demonstration videos shows the highest average success rate (88%).
>
> Additionally, we would like to note that our proposed approach is mainly focused on training the policy with hindsight-extracted trajectories; at inference time, we believe the flexibility to interact with different types of approaches to generate trajectories (via prompted LLMs, drawings, VLMs, human videos) are all equally viable approaches. In future work, we hope that stronger statements about the "most optimal" trajectory specification method can be made.
>
> Regarding extending LLM-generated Trajectories to all tasks of Fig. 4, we agree that this would be a very interesting experiment. Prior work leveraging such an approach has found significant variance with the types of tasks that LLM-generated Trajectories can accomplish (varying with scene complexity, precision required, amount of contact, etc.) [1] [2]. We empirically also experience this, where we find near 0 success rate of IK Planners using LLM-generated Trajectories on the tasks in Fig. 4, since the quality of the proposed trajectories were so low. Thus, we did not further explore LLM-generated Trajectories with RT-Trajectory on those tasks, since the baseline was so poor, and the other trajectory generation methods seemed to work well. Nonetheless, we are optimistic that improved LLMs in the future may be able to produce better trajectories (in both 2D image space [3] as well as in 3D waypoint space), and would be excited to explore these directions in future work.
>
> References:
> - [1] "Code as Policies: Language Model Programs for Embodied Control", J. Liang et al. ICRA 2023, https://arxiv.org/abs/2209.07753
> - [2] "How to Prompt Your Robot: A PromptBook for Manipulation Skills with Code as Policies", M.G. Arenas et al. CoRL 2023 TGR Workshop, https://openreview.net/forum?id=T8AiZj1QdN
> - [3] "The Dawn of LMMs: Preliminary Explorations with GPT-4V(ision)", Z. Yang et al. Arxiv Preprint 2023, https://arxiv.org/abs/2309.17421
>
> ```
> Finally, I see the point the paper makes a generalization, but I don't understand why one cannot do a more in-depth quantitative evaluation, possibly on the same tasks as before but with a different background/scene/drawer, etc.
> ```
>
> We acknowledge that additional experiments on the same tasks but with different background/scene/drawer would have provided more in-depth quantitative evaluation of generalization. Nonetheless, we do believe that generalization is an overly broad term, and can mean robust performance under many types of distribution shifts (and combinations of distribution shifts). The main focus of this work is to demonstrate RT-Trajectory can perform tasks that language or goal conditioned end-to-end learning methods find difficult to perform. We ran more than 300 episodes of evaluation on real robots, just for section 4.2 alone, on 7 new tasks and demonstrated conclusive evidence that RT-Trajectory outperforms strong language and goal conditioned baselines on these tasks.
>
> We fully agree that studying generalization on the same set of tasks but with different background/scene/drawers are interesting areas of future work, especially in tandem with building trajectory-conditioned methods on top of high-capability models such as RT-2 [4]. We are very excited to try these additional experiments for the paper, but due to the limited time of the rebuttal period, we haven’t had the chance to run these experiments yet.
>
> References:
> - [4] "Rt-2: Vision-language-action models transfer web knowledge to robotic control.", Brohan, Anthony, et al., arXiv preprint arXiv:2307.15818 (2023).

---

### Official Review · Reviewer_Ryeg · 2023-11-01

**Soundness:** 3 good
**Presentation:** 3 good
**Contribution:** 3 good
**Rating:** 8
**Confidence:** 4

**Summary:**

The paper proposes to use a rough trajectory of the end-effector in the image space as a way of task specification. This enables more flexible task specification during inference time, including a human drawing, with a human video demonstrating the task, or from LLM output. Experiments validate the effectiveness of the method and additionally demonstrate a small amount of motion generalization.

**Strengths:**

1. The paper is well-motivated, with a simple but effective modification to task specification. The paper makes a meaningful contribution to the literature and is well executed.
2. The paper is well presented, with good writing and visuals.

**Weaknesses:**

1. Generating the trajectory itself might not be trivial. For example, in the case when a person draws the trajectory, the person needs to understand the trajectory of the end-effector (including any joint limits) and project it to a 2D space. This can become a bottleneck, especially for more difficult tasks (e.g. large-range motion, long-horizon tasks). More analysis of the robustness of the trajectory specification could strengthen the paper.

2. As mentioned in the limitation section already, the method seems to be limited to a fixed-view camera with calibration, limiting the applicability of the method.

**Questions:**

Figure 8 is confusing to me. From the left visualization, the query trajectories are well captured by the training trajectories, suggesting no extrapolation and mostly interpolation. On the other hand, the right part is difficult to understand. Can the authors clarify or change the claim accordingly?

---

> ### Author Response · Authors · 2023-11-19
> **Response to Reviewer Ryeg**
>
> Thank you for the appreciation of our work and the helpful comments. We look forward to your feedback on our response.
>
> ---
>
> > Generating the trajectory might be non-trivial and can become a bottleneck.
>
> Thank you for the valuable feedback. We acknowledge that trajectory sketches are not as easy to acquire as language instructions, but also note that we show that trajectory-conditioned policies are easier to prompt for generalizing to new robotic tasks compared to language-conditioning. We believe that as foundation models (e.g., image-generating VLMs) rapidly improve, we expect that their trajectory sketch generating capabilities will improve naturally in the future and be usable by RT-Trajectory.
>
> ---
>
> > “More analysis of the robustness of the trajectory specification could strengthen the paper.”
>
> Thank you for the valuable suggestion. We have conducted additional experiments to test the robustness of our trajectory specification by adding different types of noise to human-drawn trajectory sketches: 1) adding a global offset (5 pixels) to the entire 2D trajectory, 2) adding independently sampled local offsets (5 pixels) to individual interaction markers. For each noise type, we ran a total of 22 trials on 3 skills: Place Fruit, Fold Towel, Move within Drawer. The results are shown in the table below, where we find that RT-Trajectory is robust to mild noise.
>
> | Method & noise type          | Place Fruit (12) | Fold Towel (4) | Move within Drawer (6) | Total success (22) |
> |------------------------------|------------------|----------------|------------------------|--------------------|
> | RT-Traj 2D                   |                9 |              4 |                      5 |                 18 |
> | RT-Traj 2D + global offset   |                8 |              4 |                      4 |                 16 |
> | RT-Traj 2D + marker offset   |                9 |              4 |                      4 |                 17 |
> | RT-Traj 2.5D                 |                8 |              3 |                      4 |                 15 |
> | RT-Traj 2.5D + global offset |               10 |              3 |                      4 |                 17 |
> | RT-Traj 2.5D + marker offset |                9 |              3 |                      4 |                 16 |
>
> ---
>
> > “Limited to a fixed-view camera with calibration”
>
> We agree with this limitation of our current implementation. Nonetheless, we are optimistic that this limitation can potentially be addressed by training on cross-embodiment data with different camera configurations, which we leave to future work.
>
> ---
>
> > “Fig 8 is confusing”
>
> Thank you for pointing out the unclarity of Fig. 8. We have updated Fig. 8 in the revision, by separating it into multiple figures and improving captions for more clarity.
>
> The left side of Fig. 8 shows the 10 most similar trajectories in the training data. Note that we also show additional visualizations from other viewpoints in Fig. 13 (new), where the most similar training trajectories might look more different from query ones.
>
> We find that the trajectories of some unseen tasks can be surprisingly similar to those of seen tasks. Thus, we analyze statistics of the most similar training samples over a set of queries for unseen skills on the right side of Fig. 8. In Fig. 12 (new), query trajectories of unseen skills in general see larger Frechet distances to the most similar training trajectories, compared to query trajectories from training skills. It indicates that many proposed unseen tasks show significant differences from seen tasks w.r.t. motion.
>
> Besides, in Fig. 10 (new), we show the distribution of semantics skills of the most similar training trajectories to rollout trajectories. The unseen task placing fruit into bowl is similar to a seen task move X near Y w.r.t. motion; however, their semantics are quite different, which explains why language-conditioned policy struggles to generalize in this case. Moreover, instead of “extrapolating”, we expect RT-Trajectory to stitch seen motion segments to tackle new tasks. As a good example, RT-Trajectory manages to combine motions from Open top drawer and Place object into top drawer to tackle Move within Drawer.
>
> Furthermore, we would like to point out that some criteria are missing in the current version of our proposed motion similarity metric, e.g., the end-effector orientation and when the gripper is closed or opened. For example, RT-Trajectory can respect the trajectory conditioning for Swivel Chair and does not close the gripper during execution, while its most similar training trajectories all require at least one gripper interaction. Such a difference is not captured yet. And we leave it to future work.

---

### Official Review · Reviewer_zAA2 · 2023-11-07

**Soundness:** 3 good
**Presentation:** 4 excellent
**Contribution:** 4 excellent
**Rating:** 8
**Confidence:** 5

**Summary:**

The paper proposes to abstract robot policies via rough sketches. The paper identify sketeches as a modality that is concise but yet expressive enough to represent a task, and allow for generalisation to novel tasks. The trajectory sketch in question in projected into the view of a calibrated camera. The paper explores when the trajectories are hand-drawn, derived from demonstration videos, prompted from LLMs, and generated from image generation models.

The paper is highly novel in using sketches as the abstraction for which tasks are conditioned on, and, by extensive experimental results, show the benefits of this representation for policy learning. The method appears sound, and the reviewer also appreciates the detailed evaluation on real world robot experiments and the discussion on emergent capabilities -- the retry behaviour was particularly interesting to observe. A question that might be interesting to ponder about is: does the smoothness or other property of the trajectory affect the quality of the task conditioning? Although abstracting the task as a sketch and conditioning on it is novel. The reviewer would like to point out that there have been previous attempts at leveraging human sketches to provide robot demonstrations in a paradigm known as "diagrammatic teaching" (https://arxiv.org/abs/2309.03835; https://openreview.net/forum?id=6cUysxHoL1), and should be viewed as related work.

Overall, this paper is novel, the method is sound and convincing with a solid amount of real robot experiments. The usage of trajectory sketches as a way of specifying tasks has the potential to produce many new oportunities in robot learning.

**Strengths:**

See above

**Weaknesses:**

See above

**Questions:**

See above

---

> ### Author Response · Authors · 2023-11-18
> **Response to Reviewer zAA2**
>
> Thank you for the appreciation of our novelty and the helpful comments. We are glad to hear your feedback on our response.
>
> ---
>
> > “Does the smoothness or other properties of the trajectory affect the quality of the task conditioning?”
>
> In this work, we experimented with trajectory sketches from 4 different sources: human drawings (Fig. 4 and 13), human demonstration videos (Fig. 6), prompting LLMs with Code as Policies (Fig. 16 and 17), and text-guided image generation models (Fig. 7 and 14). These sources naturally result in trajectories which show different levels of smoothness. Trajectories from human drawings tend to be smooth, while ones from human demonstration videos are usually rough due to noise from the hand pose estimator. Trajectories from prompting LLMs are usually stiff since they are created by connecting sparse waypoints. Trajectories predicted by the text-guided image generation model are the most noisy and fuzzy. A qualitative comparison between them is updated in new Fig. 21. RT-Trajectory is able to condition on trajectories generated by distinct methods with different levels of smoothness.
>
> We compare the average curvatures of the trajectories generated by different approaches for the skill Pick, shown in the table below. To compute the curvatures of a (2D) trajectory, we first reparameterize it by an arc length of 5 pixels, compute the tangent vector at each point after arc-length reparameterization, and derive the curvature as the norm of the derivative of the tangent vector. We average curvatures over 18 trajectories generated by human drawings, human demo videos and prompting LLMs as well as 36 trajectory labels sampled from training data. We do not compute curvatures for the trajectories generated by image generation models, since their outputs are images directly rather than parameterized curves.
>
> | Method         | Hindsight labels (training) | Human drawings | Human demo videos | Prompting LLMs |
> |----------------|-----------------------------|----------------|-------------------|----------------|
> | Curvature mean | 0.084                       | 0.12           | 0.256             | 0.032          |
> | Curvature std  | 0.201                       | 0.186          | 0.399             | 0.191          |
>
> ---
>
> > Related work: diagrammatic teaching
>
> Thank you for pointing out this related work. We have updated the reference in the revision. Diagrammatic teaching features reconstructing 3D motion trajectories from multi-view user sketches while RT-Trajectory focuses on learning to condition on 2D trajectory sketches. We see many interesting future directions to combine diagrammatic teaching and RT-Trajectory, e.g., extending Ray-tracing Probabilistic Trajectory Learning (RPTL) in diagrammatic teaching to handle user-specified interaction markers, and conditioning RT-Trajectory on trajectory sketches generated by RPTL.

---

### Author Response · Authors · 2023-11-19
**Shared Response to All Reviewers**

Thank you to all reviewers for their thoughtful and helpful comments! We appreciate that Reviewer zAA2 finds our work "convincing with [...] the potential to produce many new opportunities in robot learning", that Reviewer Ryeg believes our method to be "a meaningful contribution to the literature and is well executed", and that Reviewer YmAB writes that "[representing] a trajectory as an image and conditioning a model on it is an interesting idea". Below, we summarize updates we have made to improve the clarity of our manuscript and address concerns of reviewers. The new edits are highlighted in red in the updated manuscript.

**Contextualizing our method against the literature**:
- We have added a subsection highlighting classical trajectory tracking and control theory works, which are indeed highly relevant. Notably, we contextualize our method and clarify how our contributions compare to existing works.
- In addition, we also describe how our work relates to the exciting concurrent work of diagrammatic teaching.

**Improving clarity of figures**:
- We have removed the subjective Figure 2.
- We have refined Figure 8 by separating into more focused smaller figures with additional analysis (new Figures 7, 10, 11, and 12) that analyze specific aspects of motion similarity, which aims to support our main claim of motion generalization.

**Additional experiments and analysis**:
- We have done additional quantitative comparisons between the IK planner and RT-Trajectory on two skills (Fold Towel and Open Drawer) to ensure the same set of tasks are  evaluated in Table 1.  We describe the experiments in detail in our response to Reviewer YmAB, and also in an expanded Table 1 in the manuscript.
- We have conducted additional experiments to demonstrate that RT-Trajectory is robust to controlled trajectory misspecification or noise. We describe the experiments in detail in our response to Reviewer Ryeg.
- We have analyzed the varying geometric properties of different inference time trajectory sketches we study, such as by measuring the curvature of inference trajectories. We include a qualitative comparison between them in new Figure 21. RT-Trajectory is able to handle trajectories with different levels of smoothness. We describe our analysis in detail in our response to Reviewer zAA2.

We hope we have addressed some of your questions and concerns. We look forward to your responses and additional feedback!

---

### Meta-Review · Area_Chair_zWME · 2023-12-09

**Metareview:**

Synopsis: This paper presents an approach to generalizing policies to unseen settings by taking user sketches as input guidance to adapt to the new setting. The paper show that this is an effective approach to generalization, and is also capable of taking different types of input such as image generation and goal point generation.

Strengths:
+ The paper introduces a simple, yet effective idea for generalization to novel settings.
+ The presentation is clear, and the visualization are effective at explaining the algorithm and results.
+ Good real robot results.

Weaknesses:
- As pointed out by reviewer YmAB, the paper needs to do a better job of precisely stating its claims, especially w.r.t. model-based controllers. While the paper is intended to primarily be of use to data-driven approaches, some of the claims are unqualified, and may be misinterpreted as also be applicable to model-based approaches, which is inaccurate.
- The relation to some previous work (e.g, diagrammatic teaching) could be better explained

**Justification For Why Not Higher Score:**

It's a simple idea - it won't take long to convey it. The results are good, and can be briefly summarized.

**Justification For Why Not Lower Score:**

It's an idea worth sharing and discussing.

---

### Decision · Program_Chairs · 2024-01-16

Accept (spotlight)